# Standardized on-road tests assessing fitness-to-drive in people with cognitive impairments: A systematic review

**David Bellagamba**[1][ID]*, **Line Vionnet**[1][ID], **Isabel Margot-Cattin**[1], **Paul Vaucher**[2,3][ID]

1 Department of Occupational Therapy, School of Social Work & Health Sciences, HETSL, HES-SO University of Applied Sciences and Arts Western Switzerland, Lausanne, Switzerland, 2 School of Health Sciences Fribourg, University of Applied Sciences and Arts Western Switzerland (HES-SO), Fribourg, Switzerland, 3 Unit of Traffic Medicine and Psychology, University Center for Legal Medicine, Lausanne University Hospital, Lausanne, Switzerland

☯ These authors contributed equally to this work.
* Bellagamba.david@gmail.com

**Data Availability Statement:** All data are now available on Zenodo (https://zenodo.org/record/3687014#.XqxMoqgzZPY).

## Abstract

### Objective

The on-road assessment is the gold standard because of its ecological validity. Yet existing instruments are heterogeneous and little is known about their psychometric properties. This study identified existing on-road assessment instruments and extracted data on psychometric properties and usability in clinical settings.

### Method

A systematic review identified studies evaluating standardized on-road evaluation instruments adapted for people with cognitive impairment. Published articles were searched on PubMed, CINHAL, PsycINFO, Web of Science, and ScienceDirect. Study quality and the level of evidence were assessed using the COSMIN checklist. The collected data were synthetized using a narrative approach. Usability was subjectively assessed for each instrument by extracting information on acceptability, access, cost, and training.

### Results

The review identified 18 published studies between 1994 and 2016 that investigated 12 different on-road evaluation instruments: the Performance-Based Driving Evaluation, the Washington University Road Test, the New Haven, the Test Ride for Practical Fitness to Drive, the Rhode Island Road Test, the Sum of Manoeuvres Score, the Performance Analysis of Driving Ability, the Composite Driving Assessment Scale, the Nottingham Neurological Driving Assessment, the Driving Observation Schedule, the Record of Driving Errors, and the Western University's On-road Assessment. Participants were mainly male (64%), between 48 and 80 years old, and had a broad variety of cognitive disorders. Most instruments showed reasonable psychometric values for internal consistency, criterion validity, and reliability. However, the level of evidence was poor to support any of the instruments given the low number of studies for each.

**Funding:** The authors received no specific funding for this work.

**Competing interests:** The authors have declared that no competing interests exist.

## Conclusion

Despite the social and health consequences of decisions taken using these instruments, little is known about the value of a single evaluation and the ability of instruments to identify expected changes. None of the identified on-road evaluation instruments seem currently adapted for clinical settings targeting rehabilitation and occupational priorities rather than road security alone.

## Study registration

PROSPERO registration number CRD42018103276.

## Introduction

Societal changes such as urbanization, improved transport infrastructures, and increase in offers of geographically concentrated activities and commodities [1] has raised the need for effective and safe community mobility. Community mobility supports social participation [1, 2] and out-of-home activities, thus maintaining a sense of belonging, connections, and social roles [3] for community-dwelling older adults. In developed countries, motorized individualized vehicles remain the mean of transport used by most of the population [4]. Driving is also valued for its sense of independence, freedom, and competence [5, 6]. An ageing population and an increasing incidence of chronic conditions lead to more and more people living with physical, perceptual, or cognitive deficits where fitness-to-drive is compromised [7]. This growing population meets security challenges that might result in driving cessation. However, driving cessation can be seen as a health issue in itself [8], leading to identity changes (role changes), decreased self-esteem and sense of control [9], a reduction of outside-home activities [10], social isolation [11], and loss of independence [12]. A meta-analysis by Chihuri et al. [12] showed that driving cessation doubled the risks of depressive symptoms (OR = 1.91, 95% CI 1.61–2.27). Driver status is also considered as a predictor of mortality and entry into long-term care facilities among older adults [13; 14]. Finally, the transportation needs resulting from driving cessation are a predictor of caregivers burden [15]. Therefore, it is important to rely on accurate and equitable methods to assess fitness-to-drive [16] and to provide support for mobility transition for those who will face driving cessation [17].

Due to multiple interactions between individuals and their environment, driving is a complex activity [7]. It involves physical, cognitive, and perceptual (proprioceptive, visual, and auditory) skills that require fast cognitive processing and decision-making, as well as appropriate metacognition abilities to compensate difficulties [7; 18]. Michon's model breaks down cognitive load requirements for driving [19]. The strategic level requires abilities for trip planning as well as individual perceptions related to comfort. Cognitive processing is high and time pressure is low. Tactical level refers skills for maneuvering the vehicle, such as managing right-of-way, overtaking, or accepting gaps. The cognitive load is less important than at the strategic level, but the time pressure is higher. Finally, the operational level has the lowest cognitive demand but the highest time pressure for managing automated tasks (e.g., motor coordination or visual search) [19].

Health deficits affecting driving performance, such as dementia or stroke, require specific assessment approaches. While the impact of physical disorders on driving ability is often easier to compensate with vehicle adaptation, cognitive deficits are a challenge when assessing

fitness-to-drive [20]. Furthermore, general practitioners mention cognitive disorders as the first reason to justify their decision to withdraw the driver's license in 64% of cases [21], without always documenting the true effects of disorders on driving skills.

Different means are available to assess fitness-to-drive: off-road tests of prerequisites (physical, perceptual, and cognitive), simulator-based assessments, and on-road assessments. In the presence of cognitive impairments, neuropsychological tests are not considered a good predictor of fitness-to-drive [22]. It is thus advisable not to base decision-making solely on off-road testing, but also to conduct an assessment of on-road driving performance in case of doubt [16].

For this purpose, driving simulators can be used. This technology standardizes the procedure and controls a number of variables (e.g., traffic density and user behavior) [23]. This method also allows avoiding risky situations in traffic [24]. However, it is costly, and its acceptability to older adults is questionable (sickness simulation and lack of users' familiarity with technology) [24, 25]. The use of the simulator is also questionable in case of visual-perceptive disorders due to the two-dimensional representation of driving [18]. Finally, the ecological validity of simulators is questionable [26].

In terms of ecological validity, the on-road assessment is therefore considered the gold standard [27]. This assessment usually includes a closed course (e.g., in a parking lot) that allows the operational level to be safely assessed before entering traffic [18]. Depending on the construction of the instrument, the open road evaluation then allows investigation of the three levels of skills and control from Michon's model [28]. However, many variables influence driving, including weather conditions, users' behavior, road conditions, etc. The inability to control some variables makes it difficult to standardize this type of evaluation [16]. Psychometric and clinimetric values of on-road assessment methods need to be accounted for. To our knowledge, there are no systematic reviews investigating the added value of on-road assessments on the clinical decisional process for recommending driving cessation. Before starting the review, systematic review protocols on this subject were searched on PROSPERO database. No ongoing studies were found.

## Method

The aim of this systematic review is to identify and describe psychometric and clinimetric values for existing standardized on-road tests adapted for people with cognitive impairment due to acquired brain injury, dementia, or age-related disorders. The review also aims to describe costs, training requirements, accessibility, and usability of each instrument for future practical implementation. A protocol was recorded on PROSPERO (registration number CRD42018103276).

### Selection criteria

Articles meeting the following criteria were included: (a) assessments used with people with suspected or objective cognitive impairment related to acquired brain injury, dementia, or age; (b) on-road assessment; (c) standardized instruments; (d) original articles including a form of validation of the assessment (identifiable in the title or abstract); (e) articles written in English or French.

Excluded were articles comprising (a) a simulator based assessment; (b) a first driving license test; (c) the use of highly specialized equipment (high costs, low reproducibility, etc.).

### Search strategy

The search for articles was conducted on the following databases: PubMed, CINAHL Complete via EBSCO, PsycINFO via Ovid, Web of Science (core collection), and ScienceDirect

(inception—January 2018). Grey literature has not been explored, as this study focuses on available on-road assessments.

A pilot phase was carried out on PubMed to refine the equation until it could identify already known articles [29]. Three categories of keywords have been defined (assessment, driving, and cognitive impairment). To foster a conservative search, the category "cognitive disorders" was not systematically used. No additional limits were used. The same literature search was conducted at the end of the data extraction phase (January 2019) on PubMed, CINHAL and Web of Science. Full search is available at the following link: http://doi.org/10.5281/zenodo.3687014.

References of the selected articles were consulted (search backward) and the articles citing those selected were screened (search forward) for additional studies [30]. Authors of the selected articles were contacted for additional papers.

## Study selection

Titles have been simultaneously screened by two reviewers (LV and DB) against the inclusion criteria. The abstracts were then independently screened by two reviewers (LV and DB). Following the removal of duplicates, the full-text articles were assessed for eligibility against selection criteria. Any disagreements that arose between the reviewers at each stage of the study selection process have been resolved through discussion or with a third reviewer (PV). Reasons for exclusion of full-text articles were documented.

## Data extraction

Data extraction forms were created by two authors (LV and DB) and validated by a third author (PV). The extracted data include characteristics of the selected studies, characteristics of each identified on-road assessment, their psychometric and clinimetric properties, and characteristics for implementation.

The type of extracted data was defined by COnsensus-based Standards for the selection of health Measurment Instruments (COSMIN) manual [31] and included authors and year of publication, participant characteristics (number, age, gender, health condition, eligibility criteria), context, description of the test (materials, evaluators and process), and participant scores. The characteristics of the tests include: the name of the test and citation of the studies, the target population, the distance and duration of the course, the design of the road (closed and/or open road and difficulty), the items (number, categories and description), the rating system, the availability of a cut-off score, changes made between several versions, and the available versions (language) [31–33]. The psychometric and clinimetric properties of the identified on-road assessments were extracted according to the COSMIN taxonomy [34]. Finally, the implementation characteristics were cost, accessibility (i.e. how to obtain them), prerequisites (e.g. training required), and acceptability [31]. The latter represents the users' perspective on the relevance of the content and context of the test [35].

Data were extracted by two independent reviewers (LV and DB) using the standardized data extraction forms after a pilot phase on four articles. Any disagreements that arose between the reviewers during the data extraction process have been resolved through discussion or with a third reviewer (PV). Authors of selected articles have been contacted to request missing or additional data for clarification, when required.

## Assessment of the risk of bias and the quality of evidence

Assessments of the risk of bias and the quality of evidence in the selected studies were conducted using the COSMIN checklist. Although this tool was originally developed for

systematic reviews of Patient-Reported Outcome Measures (PROM), it can be used for Clinician-Reported Outcome Measures (ClinROM) [31, 36]. The evaluation of the quality of the studies was carried out in three stages: (a) assessment of the risk of bias by psychometric property by article; (b) assessment of the risk of bias by psychometric property by instrument (if more than one article); and (c) assessment of the quality of evidence by psychometric property by instrument, using a Grading of Recommendations Assessment, Development and Evaluation (GRADE) approach adapted by COSMIN. These three steps were carried out jointly by two authors (LV and DB). A third author (PV) reviewed these evaluations.

### Data analysis

It was anticipated that the heterogeneity of the identified on-road assessments would make it difficult to group the data for a meta-analysis. A narrative synthesis supported by tables was therefore carried out [37–39].

### Assessment of publication bias

Assessing the risk of publication bias in studies on measurement properties is complicated because of a lack of registry for such studies [31]. Thus, the assessment of the risk of publication bias was not carried out.

## Results

The literature search identified 5,463 records (Fig 1). Following the titles and abstracts screening and the removal of duplicates, 64 full-text articles were retrieved. Following additional search strategies (search backward, search forward, and emails to authors) 28 full-text articles were added. Thus, 92 full-text articles were assessed for eligibility and 18 met selection criteria. No additional studies were selected after the last literature search in January 2019. The study selection process is presented in the PRISMA flow diagram in Fig 1 [40].

Of these 18 articles, 12 different on-road tests were identified: the Performance-Based Driving Evaluation (PBDE) [41], the Washington University Road Test (WURT) [42], the New Haven [43], the Test Ride for Practical Fitness to Drive (TRIP) [44, 45], the Rhode Island Road Test (RIRT) [46, 47], the Sum of Manoeuvres Score (SMS) [48, 49], the Performance Analysis of Driving Ability (P-Drive) [25, 50–52], the Composite Driving Assessment Scale (CDAS) [47], the Nottingham Neurological Driving Assessment (NNDA) [53], the Driving Observation Schedule (DOS) [54], the Record of Driving Errors (RODE) [55], and the Western University's On-road Assessment (UWO) [56, 57]. Their name, the citations of the related articles, and a summary of their characteristics are in Table 1.

All but two evaluation instruments have been developed for English-speaking countries. The articles were published between 1994 and 2016. The target populations are mainly people with dementia (5 of 12) and people aged 60 and over with different levels of cognitive functioning (4 of 12). One assessment tool was intended for people who have had a stroke, one for people with multiple sclerosis, and finally one for people with cognitive disorders of various etiologies. All tests take place on open roads and six of them start with a closed course. The route is predefined for ten on-road assessments and two take place in the participants' ecological environment (DOS and CDAS). The test lasts between 45 and 60 minutes except for the DOS and the CDAS (31 minutes and 4 hours filmed over two weeks respectively) [47, 54]. This information is not available for two tests. The distance covered during the tests averaged 23.5 kilometers (ET = 10.5; 9.6–40) for nine of them. Three data were missing. A cut-off score is available for two instruments: It allows a dichotomization for the SMS [49] and a

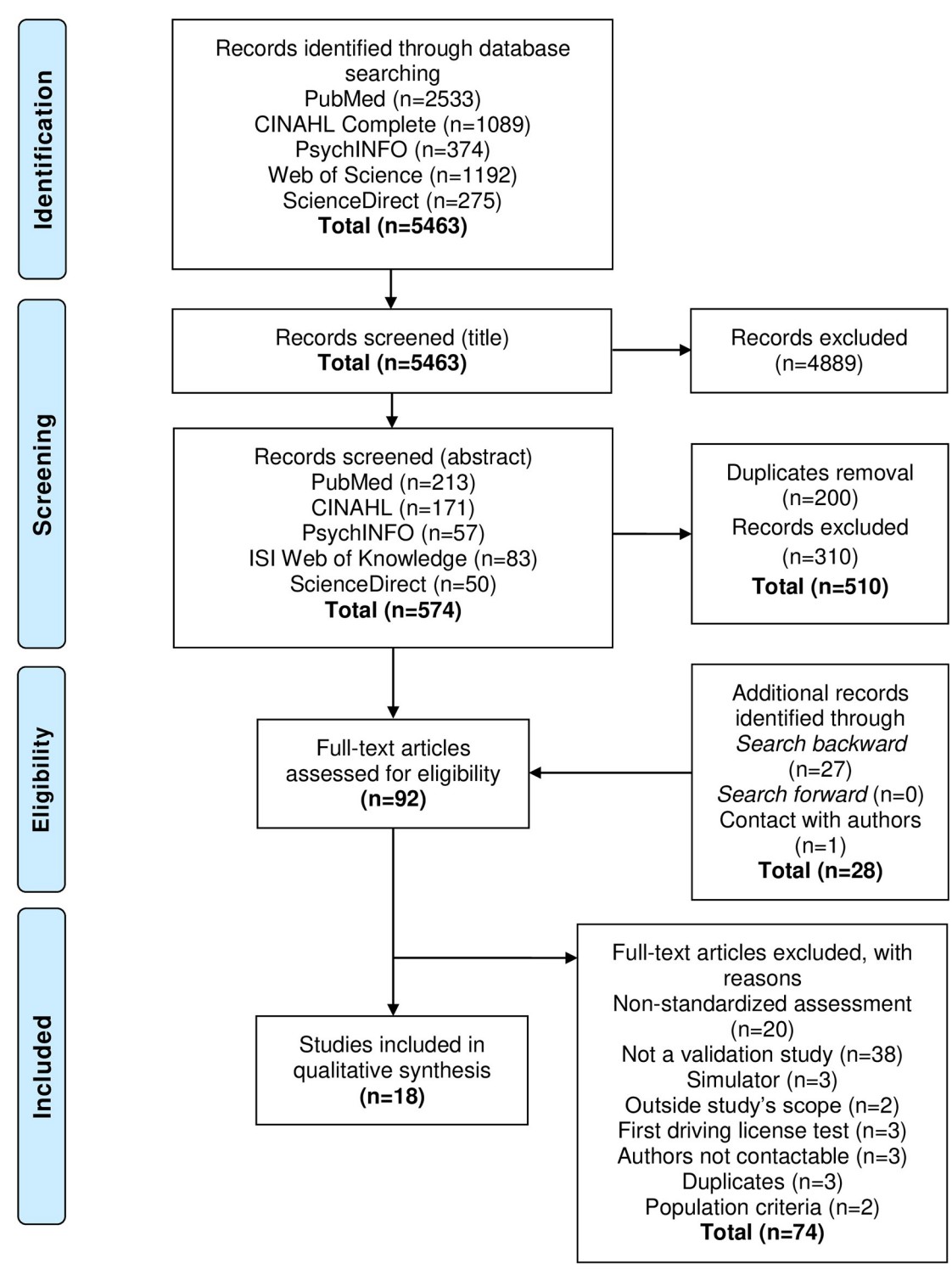

**Fig 1. Flow chart for selected studies.**

**Table 1. Synthesis of available on-road tests with characteristics.**

| On-road test (citation(s)) | Target population | Distance and/or duration | Route design | Vehicle | Cut-off score |
|---|---|---|---|---|---|
| **Performance-Based Driving Evaluation (PBDE)** Odenheimer *et al.* (1994) | People aged 60 and over with a broad range of cognitive skills | 16km (10 miles) about 45min | Standardized route (closed and open) with progressive difficulty | Dual brake vehicle | ND |
| **Washington University Road Test (WURT)** Hunt *et al.* (1997) | People with very mild or mild dementia (CDR = 0.5 or 1) | 9.6km | Standardized route (closed and open) with progressive difficulty | Dual brake vehicle with automatic gearbox | ND |
| **New Haven** Richardson and Marottoli (2003) | People aged 72 and over with a broad range of cognitive skills | 32km (20 miles) 45-60min | Standardized route (closed and open) with progressive difficulty | Dual brake vehicle | ND |
| **Test Ride for Investigating Practical Fitness to Drive: Belgian Version (TRIP)** Akinwuntan *et al.* (2003) Akinwuntan *et al.* (2005) | People with sequelae of stroke | 17-20km 45-60min | Standardized route (open) with progressive difficulty | Dual brake vehicle with automatic gearbox | ND |
| **Rhode Island Road Test (RIRT)** Brown *et al.* (2005) Ott *et al.* (2012) | People with very mild or mild dementia (CDR = 0.5 or 1) | ND | Standardized route (closed and open) with progressive difficulty | Dual brake vehicle[47] | ND |
| **Sum of Maneuvers Score (SMS)** Justiss *et al.* (2006) Shechtman *et al.* (2010) | People aged 65 and over (no exclusion criteria based on cognitive impairments) | 24km (15 miles) M = 52min | Standardized route (open) with progressive difficulty | Dual brake vehicle (with automatic gearbox[49]) | <230: unfit ≥230: fit |
| **Performance Analysis of Driving Ability (P-Drive)** Patomella and Bundy (2015) Patomella *et al.* (2010) Selander *et al.* (2011) Vaucher *et al.* (2015) | People with a neurological condition (CVA, MCI, dementia, TBI, brain tumor) | About 40km 60min | Standardized route (closed and open) with progressive difficulty | Dual brake vehicle[25,51], participant's private vehicle[52] | <81: unfit 81–85: doubtful >85: fit |
| **Composite Driving Assessment Scale (CDAS)** Ott *et al.* (2012) | People with very mild or mild dementia (CDR = 0.5 or 1) | At least 4 hours of recorded video | Participants' ecological environment following their routine | Participants' private vehicle | ND |
| **Nottingham Neurological Driving Assessment (NNDA)** Lincoln *et al.* (2012) | People with dementia | About 40min | Standardized route (open) with progressive difficulty | Choice between dual brake vehicle and private vehicle | ND |
| **Driving Observation Schedule (DOS)** Vlahodimitrakou *et al.* (2013) | People aged 75 and over (no exclusion criteria based on cognitive impairments) | M = 13.8km M = 31min 30sec | Non-standardized route starting at the participants' home and continuing on roads familiar to and chosen by participants (up to four locations) | Participants' private vehicle | ND |
| **Record of Driving Errors (RODE)** Barco *et al.* (2015) | People with dementia (CDR≥1) | 13 miles (21km) About 60min | Standardized route (closed and open) with progressive difficulty | Dual brake vehicle | ND |
| **Western University's on-road assessment (UWO)** Classen *et al.* (2016a) Classen *et al.* (2016b) | ND | 23 miles (36.2km) About 60min | Standardized route (closed and open) with progressive difficulty including a strategic driving exercise | Dual brake vehicle | ND |

ND: no data; CDR: clinical dementia rating; CVA: cerebrovascular accident; MCI: mild cognitive impairment; TBI: traumatic brain injury; M: mean

trichotomization for the P-Drive [50]. Details of items and scoring system are available on Zenodo (http://doi.org/10.5281/zenodo.3687014) under Appendix 1 (Items and scoring system).

The implementation characteristics (Table 2), with the exception of the acceptability of two on-road assessments [41, 54], required contact with the authors. Data for five instruments are not available in the absence of a response from the authors. Two instruments are available on

**Table 2. Summary table of implementation characteristics for identified on-road evaluation instruments.**

| On-road test | Acceptability | Accessibility | Cost | Prerequisite / training |
|---|---|---|---|---|
| Performance-Based Driving Evaluation (PBDE) | Acceptability of the tasks and the weather conditions piloted on 9 volunteers[41] | No response from the author | No response from the author | No response from the author |
| Washington University Road Test (WURT) | ND | Available on the internet (https://one.nhtsa.gov/people/injury/olddrive/safe/01c02.htm) | Open access | To be used by OTs, no specific training other than following the guidelines of the assessment |
| New Haven | No response from the author | No response from the author | No response from the author | No response from the author |
| Test Ride for Investigating Practical Fitness to Drive: Belgian Version (TRIP) | ND | Contact with the author | Not determined | No prerequisite but more efficiently used by driving assessment experts |
| Rhode Island Road Test (RIRT) | ND | No response from the author | No response from the author | No response from the author |
| Sum of Maneuvers Score (SMS) | ND | No response from the author | No response from the author | No response from the author |
| Performance Analysis of Driving Ability (P-Drive) | ND | Available after a 3-days training | Free of charge after the training | 3-days training in Norway (Scandinavian language) Training's cost: 800 euros |
| Composite Driving Assessment Scale (CDAS) | ND | Available in the journal Human Factors | Open access | No formal guidelines, training and qualification in the administration of road test driving assessment |
| Nottingham Neurological Driving Assessment (NNDA) | ND | Available on the internet (http://softwarelibrary.nottingham.ac.uk/medicine/about/rehabilitationageing/publishedassessments.aspx) | Open access | To be used by driving instructors specialized in the assessment of disabled drivers. A training video is available on the website. |
| Driving Observation Schedule (DOS) | Post-drive survey to assess drivers' perceptions of their DOS experience (difficulty of the tasks compared to their everyday driving, familiarity with the route, level of comfort with being observed)  **Difficulty**: DOS as about the same the difficulty (82%), as little less difficult (12%) and as little more difficult (6%)  **Familiarity**: highly familiar with the DOS route (97%)  **Comfort**: completely at ease (82%) and at ease (18%) with being observed[54] | No response from the author | No response from the author | No response from the author |
| Record of Driving Errors (RODE) | ND | Not determined | Not determined but low or no cost anticipated | To be used by OTs specialized in driving rehabilitation, online training in development |
| Western University's (UWO) on-road assessment | ND | Only used in research | Not applicable (only in research) | Training of driving rehabilitation specialist recommended |

ND: no data; OTs: occupational therapists

the Internet (WURT and NNDA), one by contact with the author (TRIP), one after training (P-Drive) and one in a journal (CDAS). Access has not yet been specified for the remaining instruments. Three on-road assessments are free of charge (WURT, CDAS, and NNDA). The price has not been established for the TRIP and RODE, and is 800€ for the P-Drive (training included). An online training is being developed for another on-road test (RODE). Finally, it is recommended to have training as a specialist in driving rehabilitation for five on-road assessments (TRIP, CDAS, NNDA, RODE, and UWO) and as an occupational therapist for one (WURT).

The characteristics of the selected studies are presented in Table 3. Participants, whose average proportion of men is 64.28% (ET = 14.5; 40–87%), are on average 70.8 years old (ET = 8.7; 48–80.2). The average study sample size is 66 (SD = 50; 6–205). A dual control vehicle was used in nine studies, without specifying the type of gearbox. A dual control vehicle with automatic transmission was used in five studies. Participants were given a choice between their private vehicle and a dual control vehicle in one study. Three studies opted for the use of participants' vehicles. Information is missing in two studies.

Table 4 lists the psychometric properties investigated in the included articles: In addition to structural validity, internal consistency, inter-rater reliability, criterion validity, and construct validity, properties related to reassessment such as responsiveness or test-retest reliability were only poorly investigated (none and twice respectively). These psychometric properties were assessed for risk of bias using the COSMIN checklist and a detailed assessment of the quality of the evidence using the GRADE approach. Details of these assessments are available on Zenodo (http://doi.org/10.5281/zenodo.3687014) under Appendix 2 (Risk of bias) and Appendix 3 (Summary of Findings). Fig 2 illustrates the quality of the evidence of psychometric properties by assessment (colors) in a synthetic way, as well as the evaluation of the aggregated results by psychometric property against the criteria for good measurement properties defined in the COSMIN checklist (+, -, ±, ?) [38]. The aggregate results are sufficient for 18 psychometric properties, undetermined for eight, insufficient for three, and inconsistent for one. The inconsistency of the criterion validity for the P-Drive can be explained by the fact that the three studies exploring criterion validity include different target populations and different expertise level of the evaluators. No quality of evidence of a psychometric property was rated as high (A), four were rated as moderate (B), five as low (C) and 21 as very low (D). This low quality is partly explained by the sample size (-1 to the quality of the evidence if $100 \geq n \geq 50$ and -2 if $n < 50$) and by the indirectness. The latter refers to a different target population than the one of this systematic review. Indeed, most tests, except the P-Drive, have a relatively narrow target population.

## Discussion

In this systematic review, 12 on-road tests were identified to assess the fitness-to-drive in people with cognitive impairment. These on-road tests are heterogeneous with regard to their components and few have been the subject of several validation studies. However, no test seems to really stand out from the others in terms of the quality of the evidence, particularly because of limited sample sizes. It is important to consider that this concerns all research on driving, because of the high costs in this field: Statistical power suffers from limited sample sizes [58].

As health conditions are potentially progressive, it is important not to decide on fitness-to-drive on a single assessment [20]. Health conditions can be transitory, episodic, or permanent. To carry out this reassessment, the test used should have good psychometric properties in terms of test-retest reliability, responsiveness, measurement stability over time, and change

**Table 3. Description of studied populations and health conditions.**

| On-road test | Citation | Country | Eligibility criteria | Sample Size | Age (years) | Gender | Health conditions |
|---|---|---|---|---|---|---|---|
| Performance-Based Driving Evaluation (PBDE) | Odenheimer et al. (1994) | United States | IC: (1) licensed drivers and (2) over the age of 60 EC: (1) major functional impairments (pain or weakness) requiring vehicular adaptations and (2) corrected static visual acuity worse than or equal to 20/200 | n = 30 | M = 72.2 / 61–89 | ♂: n = 26 (87%); ♀: n = 4 (13%) | Alzheimer's dementia (n = 3), vascular dementia (n = 3), no diagnosis (n = 24) / MMSE score: M = 26.2; SD = 5.7; 4–30 |
| Washington University Road Test (WURT) | Hunt et al. (1997) | United States | IC: (1) to be currently driving (2) possess a valid driver's license (3) have driving experiences of at least 10 years (4) have an available collateral source who was familiar with the subject's driving history and (5) have visual acuity > 20/50 | **Healthy participants (CDR = 0) n = 58**; **CDR = 0.5 group** n = 36; **CDR = 1 group** n = 29; **Total n = 123** | **Healthy participants (CDR = 0)** M = 76.8; SD = 8.6; **CDR = 0.5 group** M = 74.2; SD = 7.6; **CDR = 1 group** M = 73.1; SD = 8.2 | **Healthy participants (CDR = 0)** ♂52%; ♀48%; **CDR = 0.5 group** ♂77%; ♀23%; **CDR = 1 group** ♂50%; ♀50% | **Healthy participants (CDR = 0)** No dementia diagnosis; **CDR = 0.5 group** Very mild dementia; **CDR = 1 group** Mild dementia |
| New Haven | Richardson and Marottoli (2003) | United States | IC: (1) speak English, Spanish, or Italian (2) follow simple commands and (3) walk across a room without human assistance | n = 35 | M = 80.2; SD = 3.0 | ♂24 (68.6%); ♀11 (31.4%) | MMSE: M = 27.6; SD = 2.2; 19–30 |
| Test Ride for Investigating Practical Fitness to Drive: Belgian Version (TRIP) | Akinwuntan et al. (2003) | Belgium | ND | n = 27 | M = 60.0; SD = 13.6 | ♂22; ♀5 | CVA, right-sided brain lesion (n = 12), left-sided brain lesion (n = 15) Ischemic CVA (n = 9) and hemorrhagic CVA (n = 18) Time post-CVA (months): M = 14; SD = 8.5 |
| | Akinwuntan et al. (2005) | Belgium | ND | n = 38 | M = 53.9; SD = 12.8 | ♂31; ♀7 | First CVA, right-sided brain lesion (n = 20), left-sided brain lesion (n = 16), bilateral lesion (n = 2) ischemic CVA (n = 26) and hemorrhagic CVA (n = 12) Time post-CVA (weeks): 6–15 |
| Rhode Island Road Test (RIRT) | Brown et al. (2005) | United States | IC: (1) aged 40 to 90 (2) English speaking (3) currently driving at least one trip per week (4) have a valid driver's license and (5) have a family member willing to participate as an informant (informants had to spend time with the participants more than once weekly and to accompany the participant while driving at least once monthly during the preceding 12 months) / EC: (1) reversible causes of dementia (2) other physical, ophthalmologic, or neurological disorders that might impair driving abilities and (3) psychiatric disorders (depression not exclusionary if controlled with medications at least 6 weeks before entry into the study) | **Healthy participants (CDR = 0) n = 25**; **CDR = 0.5 group** n = 33; **CDR = 1 group** n = 17; **Total n = 75** | **Healthy participants (CDR = 0)** M = 72.4; SD = 10.2; **CDR = 0.5 group** M = 77.1; SD = 5.3; **CDR = 1 group** M = 73.2; SD = 8.3 | **Healthy participants (CDR = 0)** ♂10; ♀15; **CDR = 0.5 group** ♂21; ♀12; **CDR = 1 group** ♂10; ♀7 | **Healthy participants (CDR = 0)** No dementia diagnosis; **CDR = 0.5 group** Very mild dementia; **CDR = 1 group** Mild dementia |

(Continued)

Table 3. (Continued)

| On-road test | Citation | Country | Eligibility criteria | Sample Size | Age (years) | Gender | Health conditions |
|---|---|---|---|---|---|---|---|
| | Ott, Papandonatos, Davis and Barco (2012) | United States | IC: (1) aged 55 to 80 (2) have a valid driver's license and (3) not at-fault accidents within the past year | **Healthy participants (CDR = 0)** n = 38 | **Healthy participants (CDR = 0)** M = 68.9; SD = 7.2 | **Healthy participants (CDR = 0)** ♂60.5%; ♀39.5% | **Healthy participants (CDR = 0)** No dementia diagnosis |
| | | | For healthy participants, the following criteria are added: (4) no history of dementia and (5) MMSE score>26 | | | | |
| | | | For cognitively impaired participants, the following criteria is added: (4) CDR score = 0.5 (questionable dementia) or 1 (mild dementia) | **Cognitively impaired participants (CDR = 0.5 or 1)** n = 42 | **Cognitively impaired participants (CDR = 0.5 or 1)** M = 76.1; SD = 6.0 | **Cognitively impaired participants (CDR = 0.5 or 1)** ♂47.6%; ♀52.4% | **Cognitively impaired participants (CDR = 0.5 or 1)** Very mild dementia or mild dementia |
| | | | EC: (1) reversible causes of dementia (2) physical or ophthalmologic disorders that might impair driving abilities (3) mental retardation (4) schizophrenia (5) bipolar disorder and (6) alcohol or substance abuse within the previous year | **Total** n = 80 | **Total** M = 73.1; SD = 7.3 | **Total** ♂53.8%; ♀46.2% | |
| | | | Anxiolytic and antipsychotic medications permitted if dosages were stable for at least 6 weeks before study entry | | | | |
| Sum of Maneuvers Score (SMS) | Justiss, Mann, Stav and Velozo (2006) | United States | IC: (1) volunteers (2) possess a valid driver's license (3) older than 65 (4) minimum Snellen acuity of 30/40 and (5) having been seizure free for the past year | **Total** n = 95 | M = 75.3; SD = 6.4; 65–89 | ♂51 (54%); ♀44 (46%) | MMSE score: M = 27.2; SD = 2.3; 21–30 |
| | | | EC: (1) requiring adaptive driving equipment | **Inter-rater reliability** n = 33 | | **Inter-rater reliability** ♂56%; ♀44% | 7% of the participants <24 (presence of cognitive deficits in the sample) |
| | | | | **Test-retest reliability** n = 10 | | | |
| | Shechtman *et al.* (2010) | United States | Participants living in the community, having a valid driver's license, variety of comorbidities, but being seizure free for the past year | n = 127 | M = 74.9; SD = 6.4 | ♂68; ♀59 | No diagnosis |
| Performance Analysis of Driving Ability (P-Drive) | Patomella, Tham, Johansson and Kottorp (2010) | Sweden | IC: (1) neurological disorder (2) have a driver's license (3) have given formal consent for participation in the study | n = 205 | M = 69; 33–86 | ♂84%; ♀16% | CVA (n = 128), MCI (n = 43), dementia (n = 34) |
| | Selander, Lee, Johansson and Falkmer (2011) | Sweden | IC: (1) active drivers (minimum 3000km/year) (2) age 65 and over<br>EC: (1) fulfil physical and cognitive fit-to-drive requirements (i.e. visual problems, CVA, dementia) | n = 85 | M = 72.0; SD = 5.3; 65–85 | ♂53%; ♀47% | Participants without cognitive impairments |
| | Vaucher *et al.* (2015) | Switzerland | CI: (1) be at least 70 (2) have a valid unrestricted driver's license that is not limited geographically and (3) have their own vehicle for the on-road evaluation | n = 24 | m = 77; 70–85 | ♂66.6%; ♀33.3% | ND |
| | Patomella and Bundy (2015) | Sweden | ND | n = 99 | M = 69.3; 21–85 | ♂79%; ♀21% | CVA (n = 43), dementia (n = 34), MCI (n = 15) and others (n = 7) |

*(Continued)*

**Table 3.** (Continued)

| On-road test | Citation | Country | Eligibility criteria | Sample Size | Age (years) | Gender | Health conditions |
|---|---|---|---|---|---|---|---|
| Composite Driving Assessment Scale (CDAS) | Ott, Papandonatos, Davis and Barco (2012) | United States | IC: (1) aged 55 to 80 (2) have a valid driver's license and 3) not at-fault accidents within the past year. For the healthy participants, the following criteria are added: (4) no history of dementia and (5) MMSE score>26. For cognitively impaired participants, the following criteria is added: (4) CDR score = 0.5 or 1 EC: (1) reversible causes of dementia (2) physical or ophthalmologic disorders that might impair driving abilities (3) mental retardation (4) schizophrenia (5) bipolar disorder and (6) alcohol or substance abuse within the previous year | **Total** n = 47 | **Total** M = 71.7; SD = 7.5 | **Total** ♂55.1%; ♀44.9% | **Healthy participants (CDR = 0)** No dementia diagnosis **CDR = 0.5 group** Very mild dementia **CDR = 1 group** Mild dementia |
| | | | Anxiolytic and antipsychotic medications permitted if dosages were stable for at least 6 weeks before study entry | **Healthy participants (CDR = 0)** n = 28 | **Healthy participants (CDR = 0)** M = 68.8; SD = 7.1 | **Healthy participants (CDR = 0)** ♂67.9%; ♀32.1% | |
| | | | | **Cognitively impaired participants (CDR = 0.5 or 1)** n = 19 | **Cognitively impaired participants (CDR = 0.5 or 1)** M = 75.9; SD = 5.8 | **Cognitively impaired participants (CDR = 0.5 or 1)** ♂53.8%; ♀46.2% | |
| Nottingham Neurological Driving Assessment (NNDA) | Lincoln, Taylor and Radford (2012) | United Kingdom | ND | n = 6 | M = 78; 73–85 | ♂5; ♀1 | Diagnosis of dementia |
| Driving Observation Schedule (DOS) | Vlahodimitrakou et al. (2013) | Canada, Australia, New Zealand | IC: (1) aged 75 years and over (2) have a valid driver's license (3) drive at least 4 times per week and (4) no absolute contraindication to driving, as defined by the Austroads Fitness to Drive Guidelines | n = 33 | M = 80.1; SD = 3.4 | ♂20 (61%); ♀13 (39%) | MMSE score: M = 28.24; 25–30 |
| Record of Driving Errors (RODE) | Barco et al. (2015) | United States | IC: (1) diagnosis of dementia (2) have physician's referral for a driving assessment (3) be an active driver with current driver's license (4) Assessing Dementia-8 ≥2 by the informant (5) CDR≥1 (6) have at least 10 years of driving experience (7) have an informant available to answer questions and attend portions of the driving assessment (8) visual activity acceptable in state driving guidelines (9) speak English    EC: (1) any major chronic unstable diseases or conditions (2) severe orthopedic or musculoskeletal or neuromuscular impairments that would require adaptive equipment to drive (3) sensory or language impairments that would interfere with testing (4) use of sedating drugs and (5) a driving evaluation in the past 12 months | n = 24 | M = 69.1; SD = 9.3 | ♂17 (70.8%); ♀7 (29.2%) | Diagnoses: very mild dementia CDR = 0.5 (n = 20) and mild dementia CDR = 1 (n = 4) |

(*Continued*)

**Table 3.** (Continued)

| On-road test | Citation | Country | Eligibility criteria | Sample Size | Age (years) | Gender | Health conditions |
|---|---|---|---|---|---|---|---|
| Western University's on-road assessment (UWO) | Classen *et al.* (2016a) | Canada | **Healthy volunteer drivers** IC: (1) age between 18–64 (2) have a valid graduated driver's license and (3) drive at the time of recruitment EC: (1) be medically advised not to drive | **Total** n = 35 | **Total** M = 48; SD = 9.76 | **Total** ♂14 (40%); ♀ 21 (60%) | |
| | Classen *et al.* (2016b) | | Drivers with MS IC: (1) having MS (2) age between 18–59 (3) low physical disability (Expanded Disability Status Scale≤4) (4) cognitive impairment in 2/ 3 domains (information-processing speed, memory, or executive functions) and (5) having a valid driver's license EC: (1) having experienced a relapse within prior 3 months (2) having received a high dose of corticosteroid treatment in the month prior to testing (3) not being comfortable with driving on highways (4) not meeting the vision standards of the Ministry of Transportation (5) having taken any medications or illicit drugs that might have impacted cognition | **Healthy volunteer drivers** n = 5 | **Healthy volunteer drivers** 24–58 | **Healthy volunteer drivers** ♂2; ♀3 | **Healthy volunteer drivers** Diagnosis: without neurological disorders |
| | | | | **Drivers with MS** n = 30 | **Drivers with MS** 32–64 | **Drivers with MS** ♂12; ♀18 | **Drivers with MS** Diagnosis: multiple sclerosis (56.7% relapsing-remitting type) |

IC: inclusion criteria; EC: exclusion criteria; n: number; M: mean; SD: standard deviation; CDR: clinical dementia rating; MMSE: mini mental state evaluation; CVA: cerebrovascular accident; MCI: mild cognitive impairment; ND: no data; MS: multiple sclerosis

**Table 4. Psychometric properties of identified standardized on-road evaluation instruments.**

| On-road test | Validity | Reliability |
|---|---|---|
| Performance-Based Driving Evaluation (PBDE) (Odenheimer et al., 1994) | **Content validity** Participation of experts in driving assessment and rehabilitation in order to define the tasks to include in the test. Participation of driving experts, functional and cognitive assessment experts to develop procedures, domains to be tested, score sheets and pretesting of the road test. | **Inter-rater reliability** Correlational analyses among the raters' driving total scores Closed road: high reliability (r = 0.84) Open road: high reliability (r = 0.74) |
| | **Internal consistency** Closed road: acceptable internal consistency (α = 0.78) Open road: acceptable internal consistency (α = 0.89) | |
| | **Criterion validity** Strong significant positive correlation between instructor's global score and open road's score (r = 0.74; p<0.01) Weak significant positive correlation between instructor's global score and closed road's score (r = 0.44; p<0.05) Moderate significant positive correlation between the criterion (global score) and closed road's score, suggesting that the closed course is not an adequate method to assess on-road performance. Moderate significant positive correlation between closed road's score and open road's score (r = 0.60; p<0.01) | |
| | **Construct validity** Weak significant negative correlation between the age and the instructor's evaluation (global score) (r = -0.48; p<0.01). Strong significant positive correlation of the PBDE (open road) with the MMSE (r = 0.72; p<0.01), strong negative with complex reaction time tasks (r = -0.70; p<0.01), moderate negative with the age (r = -0.57; p<0.01), moderate positive with traffic sign recognition test (r = 0.65; p<0.01), Trail making part A (r = 0.52; p<0.01), and visual memory (r = 0.54; p<0.01) and verbal memory (r = 0.51; p<0.01). | |
| Washington University Road Test (WURT) (Hunt et al., 1997) | **Criterion validity** Criterion: instructor's global rating. Moderate significant positive correlation (τ-b = 0.60; p<0.001). Marginal to moderate significant positive correlations between the 9 subscores and the instructor's global rating (τ-b: 0.26–0.69; p<0.01) | **Inter-rater reliability** Tested with n = 10 participants. Almost perfect reliability between the instructor (global score) and the principal investigator (WURT) (κ = 0.85), almost perfect reliability between the two investigators (κ = 0.96) |
| | **Construct validity** Marginal significant negative correlation between CDR and WURT (τ-b = -0.27; p<0.001), so the more advanced the dementia is, the poorer the driving performance and vice versa (result in line with the hypothesis) | **Test-retest reliability** Tested with n = 63 participants. Global score's stability at one month for the instructor (0.53, unspecified statistics), quantitative score's stability at one month for the instructor (0.76, unspecified statistics) |
| New Haven (Richardson & Marottoli, 2003) | **Internal consistency** Acceptable internal consistency (α = 0.88) | **Inter-rater reliability** Tested with n = 357 participants. Two evaluators alternated position (front-back seat) and assessed independently the participants with the 36-item scale. For the scale: excellent reliability (ICC = 0.99) For each item: almost perfect reliability for 26 items (κ>0.91, 0.911–0.998) as well as the 10 remaining (κ>0.80) |
| | **Construct validity** Significant partial correlation coefficients (p<0.05) when controlling for distance vision between the road test score and visual attention (r = 0.43), executive functions (r = -0.38) and visual memory (r = 0.40) | |
| Test Ride for Investigating Practical Fitness to Drive: Belgian Version (TRIP) | **Criterion validity** Criterion: "pass" or "fail" category defined by the Stroke Driver Screening Assessment (SDSA) and as comparator, CARA assessor's global rating: 78.9% of agreement Comparison of the "pass" or "fail" result between the CARA assessor and the state-registered evaluator: 81.6% of agreement High significant positive correlative between TRIP's global ratings and state-registered evaluator's evaluation (r = 0.80; p<0.001) Comparison between the judgments (proportion of "pass" or "fail" people) of the CARA assessor and the judgements of the state-registered evaluator (global ratings): sensitivity of 80.6% and specificity of 100% (Akinwuntan et al., 2005) | **Inter-rater reliability** 3 databases: (1) the 27 real-life performance assessments by A (n = 17) and B (n = 10) (CARA), (2) the 27 video recordings by A (n = 10) and B (n = 17) (CARA) and (3) the 27 video recordings by C (external assessor)<br><br>**1 VS 2 (real-life performance and videos)** Subitems: level of agreement of 80% and more except for 5 subitems Items: weak to good reliability for 9/17 items (ICC: 0.42–0.85) Closed road's score: moderate reliability (ICC = 0.70) Open road's score: moderate reliability (ICC = 0.56) Global rating: moderate reliability (ICC = 0.62 and 0.64 after excluding non-reliable items)<br><br>**2 VS 3 (videos)** Subitems: level of agreement of 80% and more except for 3 subitems Items: weak to excellent reliability for 13/17 items (ICC: 0.42–1.0) Closed road's score: moderate reliability (ICC = 0.58) Open road's score: good reliability (ICC = 0.77) Global rating: good reliability (ICC = 0.80 and 0.84 after excluding non-reliable items) (Akinwuntan et al., 2003) Subitems (ordinal scale): weighted κ: 0.44–0.78. 32 subitems have a moderate to good reliability (ICC: 0.61–0.80). Items (sum of subitems): moderate to good reliability (ICC: 0.63–0.87) Global rating: good reliability (ICC = 0.83) (Akinwuntan et al., 2005) |

*(Continued)*

 

**Table 4.** (Continued)

| On-road test | Validity | Reliability |
|---|---|---|
| Rhode Island Road Test (RIRT) | **Structural validity** Factor analysis: homogeneous cluster of 21 items related to driving awareness (ICC = 0.40) that explains 31% of the variance in the scale and internal consistency (too) high ($\alpha$ = 0.93) 3 items related to stopping and parking were uninformative (items 5, 27, 28) Second cluster of 4 items related to speed control (3, 4, 13, 21) (ICC = 0.45) that explains 8% of the variance in the scale and acceptable internal consistency ($\alpha$ = 0.80) (Ott et al., 2012) | **Inter-rater reliability** Assessed with n = 20 participants. Perfect agreement for the global rating (linear weighted ratings): $\kappa$ = 0.83, quadratic weighted ratings: $\kappa$ = 0.92) High positive correlation of average RIRT score between the 2 assessors (r = 0.87) (Brown et al., 2005) |
| Sum of Maneuvers Score (SMS) | **Internal consistency** Internal consistency (too) high ($\alpha$ = 0.94), suggesting that the SMS effectively measures a single concept: the driving performance. Unidimensionality not explored (Justiss et al., 2006) | **Test-retest reliability** Interim period of one week With dichotomous scoring for each maneuver: excellent reliability (ICC = 0.91) With a score based on the 4-points scale: excellent reliability (ICC = 0.95) Low influence of the assessor's position on reliability (Justiss et al., 2006) |
| | **Criterion validity** Criterion: global rating, significant very high positive correlation between the global rating and the SMS score (r = 0.84; p<0.001) (Justiss et al., 2006) Criterion: GRS score, ROC analysis (AUC = 0.906) with a cut-off score at 230. Sensitivity = 0.91 and specificity = 0.87 (Shechtman et al., 2010) | **Inter-rater reliability** With dichotomous scoring for each maneuver: good reliability (ICC = 0.88) With a score based on the 4-points scale: excellent reliability (ICC = 0.94) Better reliability with a more detailed scale considering error's severity For the GRS: excellent reliability (ICC = 0.98) (Justiss et al., 2006) |
| Performance Analysis of Driving Ability (P-Drive) | **Structural validity** 3/27 items non-compliant with Rasch model's expectations (one generates outliers, one is over-predictable and the last one is non-compliant and needs to be revised). PCA: principal component explains 59.1% (>50%) of the variance and second component explains 4.9% (<5%); results suggest unidimensionality of the scale | **Inter-rater reliability** Excellent inter-rater reliability: random-effect intraclass correlation coefficient ICC = 0.950 (CI 95%; 0.889–0.978). Good to excellent reliability for each category (ICC: 0.875–0.963) (Vaucher et al., 2015) |
| | (Patomella et al., 2010) PCA: principal component explains 80.3% of the variance (>60%) and the variance non-explained by the first contrast is 2.4% (<5%): results suggest unidimensionality of the scale (Patomella & Bundy, 2015) | **Inter-rater reliability** Dichotomous score (pass VS marginal or fail), moderate reliability ($\kappa$ = 0.45) |
| | **Measurement invariance** Differential Item Functioning: 3 items more difficult for people with CVA VS people with MCI and one item more difficult for people with MCI VS people with CVA. Possible different item functioning between diagnoses (Patomella et al., 2010) | |
| | **Criterion validity** Criterion: subjective expert's evaluation. Optimal cut-off score at 85 (numbers concerning specificity and sensitivity not mentioned in the article, but graph available) Significant marginal positive correlation between participants' self-ratings and P-Drive evaluation ($\rho$ = 0.24; p = 0.046) (Selander et al., 2011) Significant association between P-Drive and instructors' evaluation ($R^2$ = 0.445; p = 0.021) (Vaucher et al., 2015) Criterion: global medical evaluation. ROC analysis: optimal cut-off score at 81, with a sensitivity of 0.93, a specificity of 0.92 and an AUC of 0.98 PPV = 0.95 and NPV = 0.90 (Patomella & Bundy, 2015) | |
| | **Person Separation Reliability** Person separation reliability coefficient = 0.9 (>0.7): P-Drive separates the drivers into 4 strata with person reliability = 3.06 (Patomella et al., 2010) The person separation reliability coefficient is 0.92, which indicates that P-Drive separates drivers into 4 strata (Patomella & Bundy, 2015) | |
| | **Person Response Validity** 11 participants (5%) did not demonstrate good goodness-of-fit to the model. MnSq<0.6 for 5/11 (weak threat to the Person Response Validity). Results suggest acceptable Person Response Validity (Patomella et al., 2010) 96% of the data from the occupational therapists were within acceptable range for goodness-of-fit supporting good person response validity (Patomella & Bundy, 2015) | |
| Composite Driving Assessment Scale (CDAS) (Ott et al., 2012) | **Structural validity** Factor analysis: homogeneous cluster of 20 items (ICC = 0.40) that explains 14% of the variance in the scale and acceptable internal consistency ($\alpha$ = 0.89). Second homogeneous cluster of 4 items (4, 14, 18, 25) (ICC = 0.39) that explains 12% of the variance in the scale and acceptable internal consistency ($\alpha$ = 0.73). 4 items (11, 12, 19, 26) are uninformative. | |

(Continued)

 

**Table 4.** (Continued)

| On-road test | Validity | Reliability |
|---|---|---|
| Nottingham Neurological Driving Assessment (NNDA) (Lincoln et al., 2012) | ND | **Inter-rater reliability** Perfect agreement in the overall decisions Level of agreement for the items: 100% for 7/25 items 13/25 items: discrepancies between ratings of minor errors and no error (safety not compromised in both cases) 5/25 items: discrepancies between ratings of correct or minor errors and major errors (safety compromised) Overall, discrepancies between assessors' judgements on 6/150 observations (4%) |
| Driving Observation Schedule (DOS) (Vlahodimitrakou et al., 2013) | **Face validity** Post-drive survey and GPS analysis suggest that the participants' performance on the DOS is representative of their everyday driving, supporting the DOS face validity<br><br>**Ecologic validity** Comparison between the DOS trips and participants' everyday driving with the use of a GPS over 4 months. Significant difference (p<0.0001) in terms of distance and duration. Majority of time spent on 50 and 60km/h roads during everyday driving and the DOS route. Significant difference (p<0.05) between time spent on 50km/h roads (DOS>everyday) and 80km/h roads (DOS<everyday). No difference in time driving on 40, 60, 70, 90 and 100km/h roads. Similarity between the DOS trips and participants' everyday driving supported by GPS data. Ecological validity supported by these data. | **Inter-rater reliability** Excellent reliability between observers: ICC = 0.91 (IC 95% 0.747–0.965; p<0.0001) and significant positive high positive correlation (r = 0.83; p<0.05)<br><br>**Erreur de mesure** SEM = 3% ME = 2.9% CV = 3.3% Small measures of SEM, ME and CV suggesting a high level of absolute reliability |
| Record of Driving Errors (RODE) (Barco et al. 2015) | AD | **Inter-rater reliability** Good to excellent reliability for the main categories of driving errors. For example: closed route (ICC = 0.84), low traffic (ICC = 0.90) and moderate to high traffic (ICC = 0.97). For total operational errors: ICC = 0.91. For total tactical errors: ICC = 0.95. However, low reliability for some driving errors explainable by a lower frequency of occurrence of those specific errors. For example, errors in pedal control: ICC = -0.02 |
| Western University's (UWO) on-road assessment | **Face validity** Development of the course and extraction of the main on-road components from on-road studies: involvement of a certified driving rehabilitation specialist and a certified driving instructor. These data support the UWO face validity (Classen et al., 2016a)<br><br>**Content validity** Use of a Content Validity Matrix that indicates the level of agreement (Content Validity Index) between each source (main studies on the topic of course development) and the UWO on-road course components. Several drives on the roadways to refine the course. Excellent content validity (100% agreement between the UWO on-road course and the documented on-road components identified in the literature) (Classen et al., 2016a)<br><br>**Construct validity** Tested with the known-groups method. Good construct validity: more people with MS in the fail group and more severe form of MS (progressive VS relapsing-remitting) in that group compared to the pass group (Classen et al., 2016a) | **Inter-rater reliability** Near-perfect level of agreement between the driving rehabilitation specialist and the driving instructor for the GRS by two levels (κ = 0.892; p<0.0001) and by four levels (κ = 0.952; p<0.0001) Near-perfect level of agreement for the PERS for the first type of errors (κ = 0.888; p<0.0001), for the second (κ = 0.847; p<0.0001) and for the third (κ = 0.902; p<0.0001) (Classen et al., 2016b) |

PCA: principal component analysis; ρ: Spearman correlation coefficient; p: p-value; $R^2$: determination coefficient (square of Pearson coefficient); ICC: intraclass correlation coefficient; CI: confidence interval; VPP: positive predictive value; NPV: negative predictive value; MnSq: mean-square; ROC: receiver operating characteristic; AUC: area under the curve; CVA: cerebrovascular accident; MCI: mild cognitive impairment; κ: Cohen's kappa; α: Cronbach alpha; GRS: global rating scale; PERS: priority error rating score; MS: multiple sclerosis; ND: no data; r: Pearson correlation coefficient; τ-b: Kendall rank correlation coefficient; SEM: standard error of the measurement; ME: method error; CV: coefficient of variation; CARA: Center for Determination of Fitness to Drive and Car

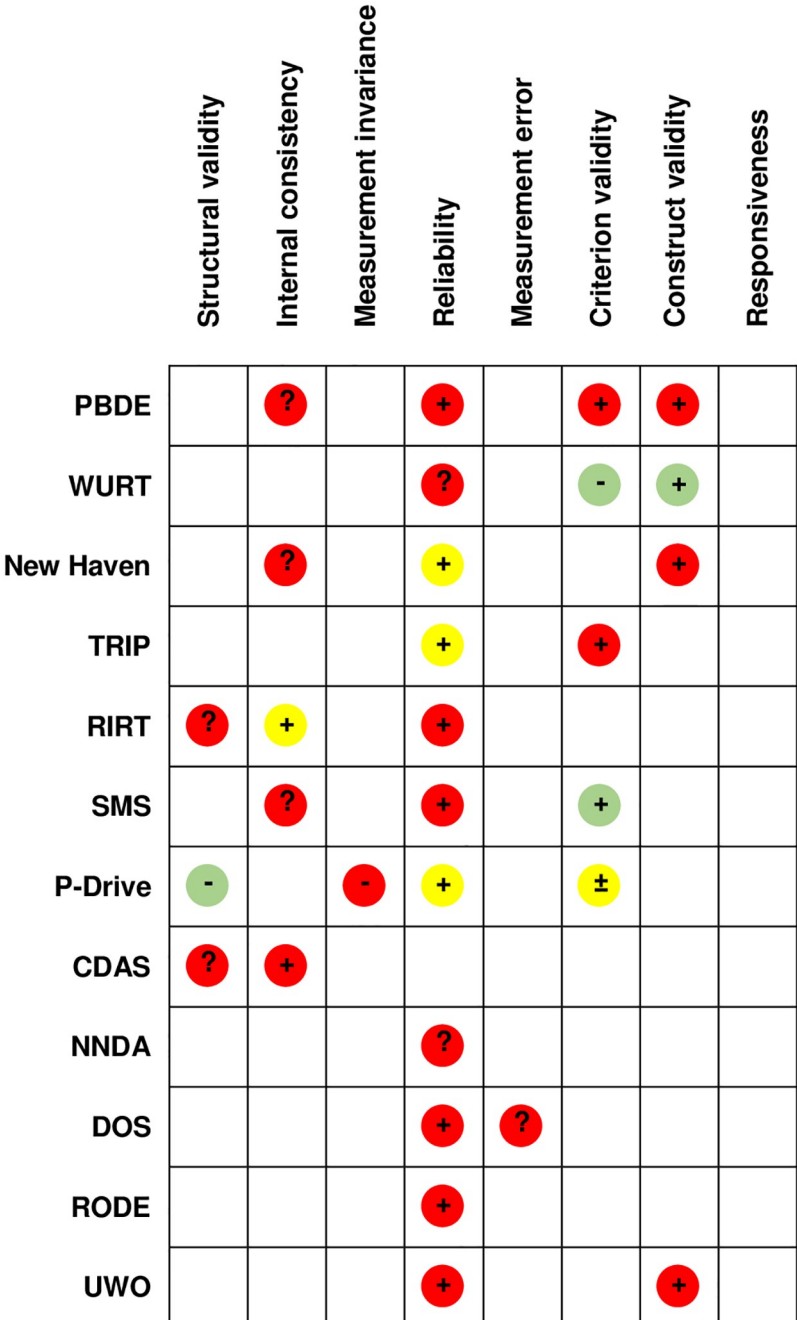

**Fig 2. Summarized results[1] and quality of the evidence[2].** [1] +: sufficient; - : insufficient; ?: indeterminate; ±: inconsistent; [2] GRADE:

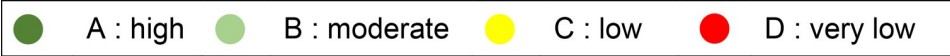

detection [59]. Among the tests identified in this systematic review, only test-retest reliability was assessed in two on-road tests (WURT and SMS) [42, 48].

On-road assessment allows for exploring the three dimensions of Michon's model: strategic, tactical, and operational [28]. Tactical and operational dimensions are systematically explored during an on-road assessment as they relate to vehicle maneuvering. However,

depending on the construction of the test, the strategic dimension is not necessarily investigated. Only UWO, CDAS, and DOS explore the strategic dimension through a task of planning of a route [47, 54, 56, 57]. It seems important to investigate all dimensions of driving and not only those related to vehicle maneuvers. Indeed, the strategic dimension influences driving safely by answering the questions of when, where, with whom, and how, among others.

Combined with reassessment, the assessment of fitness-to-drive could allow the identification of self-regulatory behaviors. These represent changes in driving behaviors (e.g., highway avoidance) that promote safe driving in cases of functional limitations, discomfort, or lack of self-confidence [60]. Self-regulation may happen in tactical, strategic, or life-goal aspects (e.g., choice of place to live in relation to one's occupation or purchase of another vehicle). Thus, the identification of self-regulatory behaviors depends on the dimensions investigated by the on-road test (tactical and strategic according to Michon's model). Since the tactical dimension is systematically explored, the identification of self-regulatory strategies at this level is accessible. This makes it possible to identify the adoption of behaviors such as avoiding distracting elements when driving (e.g., radio), or a better gap acceptance [60]. Since only three on-road tests assess the strategic dimension, work is needed to facilitate the identification of self-regulatory strategies at this level.

A cut-off score is available for two on-road tests (P-Drive and SMS) [49–51]. However, only the P-Drive allows for trichotomization of the evaluation, i.e., the classification of the people assessed in three categories (fit / doubtful as to their fitness-to-drive / unfit): Two studies were conducted and showed two different cut-off scores allowing dichotomization; it was suggested to use them as limits defining the gray zone, thus allowing a trichotomization [50]. This gray area is important as it is a gateway for interventions [32, 33]. These may be aimed at maintaining driving performance and anticipating of managing a mobility transition following driving cessation [17].

Several studies have allowed participants to use their private vehicles [47, 52, 54] or have given them a choice [53], suggesting some ecological validity. In addition, two tests are conducted in an ecological environment (CDAS and DOS) [47, 54]. According to the Person-Environment-Occupation-Performance Model (PEOP), occupational performance emerges from the interaction between the person, their environment, and the occupation [61]. The use of standardized roads and the instructors' professional vehicle ignores the potential influence of the ecological environment on performance. In this context, the ecological validity of the assessment is questionable [54]. Indeed, familiarity with the environment while driving reduces cognitive load, particularly at the attention level, as suggested in a simulator study [62]. Familiarity also promotes spatial orientation [63]. In addition, the use of the instructors' professional vehicle results in a reduction of performance [64]. However, familiarity with the road increases distractibility while driving [65] as well as reaction time [66]. In addition, risk perception and ability to respect speed regulation are reduced [67, 68]. All ages combined, the majority of accidents occur on familiar roads [69]. However, these results are from studies that are not limited to participants with cognitive impairment. Studies suggest that tactical self-regulation is influenced by the level of familiarity with the environment in which the evaluation takes place. This could raise safety issues for evaluations that do not cover unknown environments.

Furthermore, the choice of the vehicle could have an influence on safety during the test. In the event of a serious error, the absence of dual control could impede on the evaluator's ability to intervene quickly [33]. In this context, beginning the evaluation on a closed-course could allow an assessment of the operational dimension of driving beforehand. This ensures sufficient control over safety before entering traffic [28]. However, this component of the test was

not identified in all the tests [44, 45, 48, 49, 53], including those conducted in an ecological setting [47, 54].

The involvement of two people, a driving instructor to ensure safety and usually an occupational therapist for the scoring, was very common. As confirmed by other studies, only three instruments, the CDAS, the New Haven, and the RIRT relied on a single person [43, 46, 47]. However, the added value of having a second evaluator was never properly studied. The shortage of qualified occupational therapists in some countries, such as Switzerland [70], could question this practice. Resources could be optimized by having occupational therapists focus on activity analysis, community mobility, and mobility transition [71]. Given the almost perfect inter-rater reliability between occupational therapists and instructors identified in two studies [42, 57], the latter appear to represent a potential way to address the shortage of therapists in this domain. According to Nilsen [72], another resource to consider in the implementation of an evaluation instrument is the accessibility to appropriate training. One training course is available for the P-Drive in Scandinavia and another is under development for the RODE. The WURT does not require any specific training, but simply to follow the protocol.

Even though systematic review protocols were searched before our study, a systematic review concerning the reliability and validity of on-road driving tests has recently been published [73]. They identified 21 on-road tests. This difference can be explained by the different eligibility criteria: They included studies not restricted to people with cognitive impairment, and studies in which the validation process couldn't be identified in the title or abstract. The authors also found that validation studies mainly focused on the inter-rater reliability, that measurement error and responsiveness were not investigated despite their importance. However, these psychometric properties are of great importance, as explained previously and as mentioned as well by Sawada et al. [73].

Given the limitations of on-road tests concerning their psychometric properties and their components' variability, these tests are not as promising as expected and costly in terms of financial and human resources. However, it is necessary to have reliable, valid and specific instruments in order to support decision making concerning the withdrawal of the driving licence, its retention or its restriction [74]. A less expensive and potentially effective alternative would be the use of cognitive tests to assess fitness-to-drive [75]. A systematic review exploring the relationship between cognitive tests and on-road driving performance in people with dementia has been conducted [75]. It appears that composite batteries of cognitive tests are more appropriate to predict on-road driving performance than cognitive tests focused on a single cognitive ability. Though, these composite batteries are not sufficiently validated: Cut-off scores enabling trichotomization would be useful. One specific test, the Useful Field Of View (UFOV©) appears to be a potential predictor of driving fitness-to-drive [76]. This instrument assesses cognitive domains (selective attention, divided attention and processing speed) in three tasks [76]. However, there are many different versions of this instrument and some authors have modified it for their studies [77]. For this reason, the interpretation of the results is more complex than it appears. In addition, the UFOV© possibly measures visual functions that are not directly related to driving. Indeed, in addition to measuring processing speed and different types of attention, it involves visual functions such as visual acuity. The latter is not a predictor of fitness-to-drive [77]. As visual acuity decreases with age, the UFOV© score decreases and this does not necessarily reflect a poorer driving performance. Thus, it seems important to have age-related normative values for UFOV© [78].

Another alternative would be the use of simulators: They allow to evaluate complex behaviors in a controlled environment when these behaviors might not be safe, practical nor ethical during an on-road test [79]. They must be immersive, sufficiently challenging and have complex scenarios in order to emulate accurately and consistently the real-world performance

[79]. Technological advances have made it possible to make simulators much more dynamic by improving visual quality, for example, but also by being able to control traffic, include other road users and modify the driving environment in real time according to the behavior of the person being assessed [79]. These authors stress the importance of the simulator's choice, as it is widely used in practice and research and, depending on the simulator, it can lead to biased interpretation. There are few validation studies on simulators. The authors suggest that the simulator does not systematically represent a valid assessment of driving performance [79]. Each simulation setting is unique and must be valid, which can lead to a very important financial investment: Simulators are therefore much more expensive than they seem.

Finally, according to Wynne et al. [79], the perception of risk in simulators is also reduced: The engagement in risky situations is higher and compliance with traffic regulations is reduced. Simulator's driving performance is influenced by this absence of stress, which can be present during an on-road test. In addition, the simulator does not allow the evaluation of situations where familiarity with the environment could have an impact on on-road driving performance [79].

## What does this study add

The choice of components (ecological environment, private vehicle, evaluation of the strategic dimension) when constructing the instrument is of great importance as they influence performance and safety during the evaluation.

- This study brings elements regarding implementability characteristics in addition to the psychometric properties of the instruments.

- This systematic review was specifically focused in on-road tests for people with cognitive impairment with a comparison of their psychometric properties and components.

- Familiarity's influence must be explored in order to guide on-road tests' elaboration.

## Recommendations for practice and research

In conclusion, none of the methods for assessing fitness-to-drive appears to be ideal and none of them alone appears to be sufficient. Thus, these different tests could be combined in order to fuel the decision making process regarding the withdrawal, restriction or retention of the driving licence. The choice of an evaluation instrument remains an important concern: It is necessary to use valid instruments. In sum, off-road tests can be used to identify situations requiring further assessment. On-road or simulator-based performance tests could support decision making in ambiguous situations. In this regard, Sawada et al. consider WURT, P-Drive and TRIP as potential gold standards [73]. In the present study, WURT, P-Drive, TRIP, RIRT, SMS and New Haven seem to present the best results in terms of summarized results and quality of the evidence, the others being dismissed out-of-hand because of their weak results or quality of the evidence. Of these instruments, the P-Drive seems to stand out from the others: As mentioned previously, it allows trichotomization, a training is available, its target population is larger than in the other instruments and it is the most studied instrument among the articles selected in this systematic review. Nevertheless, limitations remain: The strategic dimension is not explored as well as the responsiveness, the measurement error and the test-retest reliability. When used in other languages than Swedish or English, a transcultural adaptation would be necessary to make this assessment (and the training) accessible. Finally, its implementation is limited by the shortage of occupational therapists. Specific

interdisciplinary regional training programs for occupational therapists and driving instructors could facilitate the implementation of improved methods for evaluating fitness-to-drive in people with cognitive impairment.

There seems to be an association between environmental familiarity and driving performance. However, more studies on this subject are needed to determine the relevance of conducting a driving assessment in an ecological environment at the expense of road standardization and vice versa. In addition, this association should be explored in people with cognitive impairments. In this context, there is also a need for studies to evaluate the responsiveness of evaluation instruments to known changes in health conditions, and to develop methods to distinguish driving lapses and errors due to health conditions from those due to other causes. A possible solution could be to develop instruments that rely on at least three separate driving phases: evaluation, intervention, and re-evaluation.

Finally, it is relevant to provide evidence of the added value or necessity of the involvement of occupational therapists during the on-road assessment, as that has become the norm in many countries.

## Limitations of the study

As some tests were initially developed on simulators (e.g. P-Drive), it is possible that some validation studies were not selected due to the defined selection criteria. Some data concerning the psychometric properties may therefore be missing. In addition, different steps of the systematic review were carried out jointly for educational purposes, which may limit the quality of the methodology.

As the choice of information sources is limited to five databases, it is possible that some on-road assessments available from institutional or government reports have not been included. The same applies to tests developed in doctoral theses, for example. Finally, test-retest and inter-rater reliability were not differentiated when assessing the risk of bias using the COSMIN checklist. The readability of the results may thus be affected.

## Conclusion

This systematic review identified 12 on-road evaluation instruments adapted for people with cognitive impairment. When compared with recommendations from the scientific literature, these instruments do not comply to scientific standards for medical diagnosis procedures. Following a single-step evaluation procedure, risks of falsely recommending driving cessation, which might lead to important health consequences, could still be present. This is particularly the case for most evaluation methods that do not provide the opportunity to express uncertainty in the results. Trichotomization is of great importance as it favors interventions that could help maintain driving, or preconize a transition to anticipate driving cessation and facilitate mobility transition.

## Supporting information

**S1 Protocol.**
(PDF)

**S1 Checklist. PRISMA checklist.**
(PDF)

## Acknowledgments

This article is based on the results of a Master Thesis conducted within the joint Master of Science (MSc) in Health Sciences HES-SO (University of Applied Sciences and Arts Western Switzerland) and University of Lausanne (UNIL), major in Occupational Therapy, at HES-SO Master.

## Author Contributions

**Conceptualization:** David Bellagamba, Line Vionnet, Isabel Margot-Cattin, Paul Vaucher.

**Data curation:** David Bellagamba, Line Vionnet.

**Formal analysis:** David Bellagamba, Line Vionnet.

**Investigation:** David Bellagamba, Line Vionnet.

**Methodology:** David Bellagamba, Line Vionnet, Paul Vaucher.

**Project administration:** David Bellagamba, Line Vionnet, Paul Vaucher.

**Resources:** David Bellagamba, Line Vionnet, Paul Vaucher.

**Supervision:** Isabel Margot-Cattin, Paul Vaucher.

**Validation:** David Bellagamba, Line Vionnet, Paul Vaucher.

**Visualization:** David Bellagamba, Line Vionnet.

**Writing – original draft:** David Bellagamba, Line Vionnet.

**Writing – review & editing:** David Bellagamba, Line Vionnet, Isabel Margot-Cattin, Paul Vaucher.

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
