## [Decision Letter · Decision Letter 0]

24 Feb 2020

PONE-D-19-34890

Standardized on-road tests assessing fitness-to-drive in people with cognitive impairments: a systematic review

PLOS ONE

Dear Mr Bellagamba,

Thank you for submitting your manuscript to PLOS ONE. After careful consideration, we feel that it has merit but does not fully meet PLOS ONE’s publication criteria as it currently stands. Therefore, we invite you to submit a revised version of the manuscript that addresses the points raised during the review process.

Your paper has been reviewed by one acknowledged expert in the field. Overall, the Reviewer has identified some potential in the study, highlighting its importance but also raising key concerns and questions on the added value that your manuscript may represent. My personal opinion is coherent to what the reviewer expresses in their review: more information, context, implications and benefits related to the study should be developed -and extensively discussed- in a revised version of the paper.

We would appreciate receiving your revised manuscript by Mar 30 2020 11:59PM. To enhance the reproducibility of your results, we recommend that if applicable you deposit your laboratory protocols in protocols.io, where a protocol can be assigned its own identifier (DOI) such that it can be cited independently in the future. For instructions see: http://journals.plos.org/plosone/s/submission-guidelines#loc-laboratory-protocols

We look forward to receiving your revised manuscript.

Kind regards,

Sergio A. Useche, Ph.D.

Academic Editor

PLOS ONE

Journal Requirements:

3. Thank you for including your PRISMA Checklist however we note it is part of your cover letter as opposed to uploaded as a separate supporting information file, as per our author guidelines (https://journals.plos.org/plosone/s/submission-guidelines#loc-systematic-reviews-and-meta-analyses) can you please upload the PRISMA Checklist as a supporting information file.

Reviewers' comments:

Reviewer's Responses to Questions

**Comments to the Author**

1. Is the manuscript technically sound, and do the data support the conclusions?

Reviewer #1: Partly

2. Has the statistical analysis been performed appropriately and rigorously? 

Reviewer #1: Yes

3. Have the authors made all data underlying the findings in their manuscript fully available?

Reviewer #1: Yes

4. Is the manuscript presented in an intelligible fashion and written in standard English?

Reviewer #1: Yes

5. Review Comments to the Author

Reviewer #1: This paper presents a meta-analysis of on road assessments for driving fitness. The authors review the literature and are generally unable to come to conclusions about all tests other than one. The one that looks best is extremely expensive and has a training program associated with it.

1. Since the studies are apparently of such low quality, why did the authors not compare the quality of the one acceptable assessment to results from driving simulator studies. They are dismissed out of hand, but many of these simulators have lots of data associated with them.

2. How does this paper go beyond the results of two very recent systematic reviews on the same topic?

3. What is the point of this review? If there is only one suitable on road assessment and it is delivered in Sweden, how does this advance the field? These are obviously very challenging to develop and implement.

4. The review does not measure other predictive studies predicting fitness to drive from other assessments such as Useful Field of View. Since the on-road assessment appear to be very weak, why not look to other predictive data? UFOV was shown to be improved in the ACTIVE trial and predicted better driving over the next 10 years.

6. PLOS authors have the option to publish the peer review history of their article (what does this mean?). If published, this will include your full peer review and any attached files.

Reviewer #1: No

---

## [Author Response · Author response to Decision Letter 0]

24 Mar 2020

This paper presents a meta-analysis of on road assessments for driving fitness. The authors review the literature and are generally unable to come to conclusions about all tests other than one. The one that looks best is extremely expensive and has a training program associated with it.

Response: We find that 800 euros is not extremely expensive, since the instrument is included in the price. Moreover, the fact that a training program exists is a proof of validity and inter-rater reliability.

Changes: “According to Nilsen [72], another resource to consider in the implementation of an evaluation instrument is the accessibility to appropriate training.” (p.34) 

“Specific interdisciplinary regional training programs for occupational therapists and driving instructors could facilitate the implementation of improved methods for evaluating fitness-to-drive in people with cognitive impairment.” (p.38)

We explained why we dismissed out-of-hand all but 6 tests. We then explained why we kept only the P-Drive based on recommendations (trichotomization, training, target population, study validation). Here are the changes:

“In conclusion, none of the methods for assessing fitness-to-drive appears to be ideal and none of them alone appears to be sufficient. Thus, these different tests could be combined in order to fuel the decision making process regarding the withdrawal, restriction or retention of the driving licence. The choice of an evaluation instrument remains an important concern: It is necessary to use valid instruments. In sum, off-road tests can be used to identify situations requiring further assessment. On-road or simulator-based performance tests could support decision making in ambiguous situations. In this regard, Sawada et al. consider WURT, P-Drive and TRIP as potential gold standards [73]. In the present study, WURT, P-Drive, TRIP, RIRT, SMS and New Haven seem to present the best results in terms of summarized results and quality of the evidence, the others being dismissed out-of-hand because of their weak results or quality of the evidence. Of these instruments, the P-Drive seems to stand out from the others: As mentioned previously, it allows trichotomization, a training is available, its target population is larger than in the other instruments and it is the most studied instrument among the articles selected in this systematic review. Nevertheless, limitations remain: The strategic dimension is not explored as well as the responsiveness, the measurement error and the test-retest reliability.” (p.37)

1. Since the studies are apparently of such low quality, why did the authors not compare the quality of the one acceptable assessment to results from driving simulator studies. They are dismissed out of hand, but many of these simulators have lots of data associated with them.

Response: Thank you very much for your comment. As occupational therapists, we focused our interest on ecological environment, i.e. on on-road driving performance evaluation. Your comment allows us to go further in our thinking, but not so far as it’s not the primary objective of our study.

Changes: “Another alternative would be the use of simulators: They allow to evaluate complex behaviors in a controlled environment when these behaviors might not be safe, practical nor ethical during an on-road test [79]. They must be immersive, sufficiently challenging and have complex scenarios in order to emulate accurately and consistently the real-world performance [79]. Technological advances have made it possible to make simulators much more dynamic by improving visual quality, for example, but also by being able to control traffic, include other road users and modify the driving environment in real time according to the behavior of the person being assessed [79]. These authors stress the importance of the simulator’s choice, as it is widely used in practice and research and, depending on the simulator, it can lead to biased interpretation. There are few validation studies on simulators. The authors suggest that the simulator does not systematically represent a valid assessment of driving performance [79]. Each simulation setting is unique and must be valid, which can lead to a very important financial investment: Simulators are therefore much more expensive than they seem.

Finally, according to Wynne et al. [79], the perception of risk in simulators is also reduced: The engagement in risky situations is higher and compliance with traffic regulations is reduced. Simulator’s driving performance is influenced by this absence of stress, which can be present during an on-road test. In addition, the simulator does not allow the evaluation of situations where familiarity with the environment could have an impact on on-road driving performance [79].” (p.36)

2. How does this paper go beyond the results of two very recent systematic reviews on the same topic ?

Response: Thank you for your comment. We added a section at pages 36-37. Sawada et al. only discussed the psychometric properties of the identified tests. Our study also discussed their components and identified implementability characteristics.

To our knowledge, there is not a second systematic review on on-road driving evaluations but there is one concerning simulators. If there is one, we would be interested in reading it.

Changes: “What does this study add

• The choice of components (ecological environment, private vehicle, evaluation of the strategic dimension) when constructing the instrument is of great importance as they influence performance and safety during the evaluation.

• This study brings elements regarding implementability characteristics in addition to the psychometric properties of the instruments.

• This systematic review was specifically focused in on-road tests for people with cognitive impairment with a comparison of their psychometric properties and components.

• Familiarity’s influence must be explored in order to guide on-road tests’ elaboration.”

3. What is the point of this review ? If there is only one suitable on-road assessment and it is delivered in Sweden, how does this advance the field ? These are obviously very challenging to develop and implement.

Response: Our initial concern was practical: There is no standardized on-road test in Switzerland and we are interested in implementing one. In that purpose, we needed to compare their psychometric properties and components as these tests are very heterogenous. Our recommendations can guide a choice and identify gaps of knowledge.

No change has been made specifically for this subject.

4. The review does not measure other predictive studies fitness to drive from other assessments such as Useful Field of View. Since the on-road assessment appear to be very weak, why not look to other predictive data? UFOV was shown to be improved in the ACTIVE trial and predicted better driving over the next 10 years.

Response: Thank you for your comment. We added reflections about cognitive tests, batteries of cognitive tests and UFOV©, but couldn’t really extent on this topic as we wanted to focus on on-road tests as mentioned in the aim of our study. We think that another study would be more appropriate. We tried to link the different methods of evaluation (off- and on-road).

Changes: “Given the limitations of on-road tests concerning their psychometric properties and their components’ variability, these tests are not as promising as expected and costly in terms of financial and human resources. However, it is necessary to have reliable, valid and specific instruments in order to support decision making concerning the withdrawal of the driving licence, its retention or its restriction [74]. A less expensive and potentially effective alternative would be the use of cognitive tests to assess fitness-to-drive [75]. A systematic review exploring the relationship between cognitive tests and on-road driving performance in people with dementia has been conducted [75]. It appears that composite batteries of cognitive tests are more appropriate to predict on-road driving performance than cognitive tests focused on a single cognitive ability. Though, these composite batteries are not sufficiently validated: Cut-off scores enabling trichotomization would be useful. One specific test, the Useful Field Of View (UFOV©) appears to be a potential predictor of driving fitness-to-drive [76]. This instrument assesses cognitive domains (selective attention, divided attention and processing speed) in three tasks [76]. However, there are many different versions of this instrument and some authors have modified it for their studies [77]. For this reason, the interpretation of the results is more complex than it appears. In addition, the UFOV© possibly measures visual functions that are not directly related to driving. Indeed, in addition to measuring processing speed and different types of attention, it involves visual functions such as visual acuity. The latter is not a predictor of fitness-to-drive [77]. As visual acuity decreases with age, the UFOV© score decreases and this does not necessarily reflect a poorer driving performance. Thus, it seems important to have age-related normative values for UFOV© [78].” (p.35)

---

## [Decision Letter · Decision Letter 1]

29 Apr 2020

Standardized on-road tests assessing fitness-to-drive in people with cognitive impairments: a systematic review

PONE-D-19-34890R1

Dear Dr. Bellagamba,

We are pleased to inform you that your manuscript has been judged scientifically suitable for publication and will be formally accepted for publication once it complies with all outstanding technical requirements.

With kind regards,

Sergio A. Useche, Ph.D.

Academic Editor

PLOS ONE

Additional Editor Comments (optional):

Reviewers' comments:

Reviewer's Responses to Questions

**Comments to the Author**

1. If the authors have adequately addressed your comments raised in a previous round of review and you feel that this manuscript is now acceptable for publication, you may indicate that here to bypass the “Comments to the Author” section, enter your conflict of interest statement in the “Confidential to Editor” section, and submit your "Accept" recommendation.

Reviewer #1: All comments have been addressed

2. Is the manuscript technically sound, and do the data support the conclusions?

Reviewer #1: Yes

3. Has the statistical analysis been performed appropriately and rigorously? 

Reviewer #1: Yes

4. Have the authors made all data underlying the findings in their manuscript fully available?

Reviewer #1: Yes

5. Is the manuscript presented in an intelligible fashion and written in standard English?

Reviewer #1: Yes

6. Review Comments to the Author

Reviewer #1: The authors have attended to my review comments. They have clarified a couple of minor issues I raised and have mentioned some other less desirable strategies for fitness to drive. Given the nature of the available data, the paper does not need additional revision.

7. PLOS authors have the option to publish the peer review history of their article (what does this mean?). If published, this will include your full peer review and any attached files.

Reviewer #1: No

---

## [Editor Report · Acceptance letter]

5 May 2020

PONE-D-19-34890R1 

Standardized on-road tests assessing fitness-to-drive in people with cognitive impairments: a systematic review 

Dear Dr. Bellagamba:

I am pleased to inform you that your manuscript has been deemed suitable for publication in PLOS ONE. Congratulations! Your manuscript is now with our production department. 

With kind regards,

on behalf of

Dr. Sergio A. Useche 

Academic Editor

PLOS ONE